# Polarizability matters in enantio-selection

Fumin Chen[1,2,4], Yu Chen[2,4], Xiao-Yong Chang [2], Dongxu He[2], Qingjing Yang [2], David Zhigang Wang[3], Chen Xu [2]✉, Peiyuan Yu [2]✉ & Xiangyou Xing [2]✉

The prevalence of chirality, or, handedness in biological world is a fundamental phenomenon and a characteristic hallmark of life. Thus, understanding the origin of enantio-selection, i.e., the sense and magnitude of asymmetric induction, has been a long-pursued goal in asymmetric catalysis. Herein, we demonstrated a polarizability-derived electronic effect that was shown to be capable of rationalizing a broad range of stereochemical observations made in the field of asymmetric catalysis. This effect provided a consistent enantio-control model for the prediction of major enantiomers formed in a ruthenium-catalyzed asymmetric transfer hydrogenations of ketones. Direct and quantitative linear free energy relationships between substrates' local polarizabilities and observed enantio-selectivity were also revealed in three widely known asymmetric catalytic systems covering both reductions and oxidations. This broadly applicable polarizability-based electronic effect, in conjunction with conventional wisdom mainly leveraging on steric effect considerations, should aid rational design of enantio-selective processes for better production of chiral substances.

Polarizability, as a fundamental physical property, characterizes the sensitivity of the electron density of an atom or a molecule to distortion under the influence of an external electric field[1]. The exact value of a group's polarizability often depends on its specific chemical environment. 60 years ago, Pearson laid the foundation of the classic hard and soft acids and bases (HSAB) theory[2], in which he showed that polarizability, as a basic electronic property, measures whether an acid or a base is "soft" (more polarizable) or "hard" (less polarizable), and that substances of similar polarizabilities typically react faster and form stronger bonds with each other. Later, Parr and co-workers suggested that polarizability is a parameter as universal and fundamental as energy itself, and that there may be such general rules as the principles of the maximum hardness and of the minimum polarizability dictating the reactivity and selectivity in chemical events[3]. The implication of polarizability on chemical reactivity was further highlighted by Zimmerman and co-workers in a seminal contribution in which it was shown that electronic movements, i.e., reshuffling of electronic densities, in chemical reactions precisely follow a polarizability rule, as it is always the case that the more polarizable bonds break more easily and the less

polarizable bonds form more easily[4]. However, previous studies on polarizability effects mainly focused on achiral systems, and the underlying connection between polarizability and the enantio-selection in stereochemical events has been largely overlooked in the past decades. Jacobsen and co-workers reported intriguing examples that polarizability of the arene of a thiourea catalyst strongly influences the enantioselectivity of the polycyclization of the hydroxylactams[5] and the ring-opening of episulfonium ions with indoles[6]. Schreiner and co-workers reported that, with a given catalyst in the Corey–Bakshi–Shibata (CBS) reductions, enantioselectivities increase with the computed substrates' polarizabilities per volume[7]. The trends in the above were understood based on the fact that more polarizable substrates lead to stronger non-covalent interactions (cation-π[5,6] or London dispersion[7]) with the catalyst and thus to higher enantioselectivities. Polarizability of chiral ligands has been further shown by Sigman and co-workers to influence enantio-selection in transition-metal catalyzed asymmetric transformations through multivariate correlation studies[8,9]. These results, although have been only sporadically reported, demonstrate that polarizability may have a general influence on enantioselective processes.

[1]School of Chemistry and Chemical Engineering, Harbin Institute of Technology, Harbin 150001, China. [2]Shenzhen Grubbs Institute and Department of Chemistry, Guangdong Provincial Key Laboratory of Catalysis, Southern University of Science and Technology, Shenzhen 518055, China. [3]Shenzhen Youwei Tech Group, Shenzhen 518057, China. [4]These authors contributed equally: Fumin Chen, Yu Chen. ✉e-mail: xuc@sustech.edu.cn; yupy@sustech.edu.cn; xingxy@sustech.edu.cn

The pioneering work of Parr, Pearson and Yang suggested that the local softness/hardness, i.e., local electronic polarizability that is positioned nearest to the bond-forming and -breaking events in the relevant transition state, generally serves as more effective determinants for a molecule's stability and reactivity[2,3,10-14]. Thus, in agreement with Brewster's suggestion[15], we hypothesized that the polarizability at the atoms and bonds directly attached to the forming chiral center might be particularly strongly related to enantioinduction. That is, echoing with the proposal of Parr et al.[2,3,10-14], it is the local polarizability that really matters. Recent advances in computational chemistry have made it a routine task to incorporate London dispersion interactions into the density functional theory framework (DFT-D), in which the computation of atomic polarizabilities has become readily accessible[16]. Leveraging on such remarkable development in this field, it has now become feasible for the quantitative mapping of substrates' local polarizabilities and examining their potential role in enantio-selection.

Herein, with the goal of systematically probing the role of polarizability in enantio-selection, we firstly examined the well-established Noyori-type asymmetric transfer hydrogenation as the model system. As shown in Fig. 1a, ketone substrates **A** featuring two substituents of different local polarizabilities ($R^1 > R^2$) were hydrogenated with the same bottom-face selection using a highly efficient Ru-catalyst **B** recently developed by us[17-20]. Good to high ees were generally recorded. In each case, the local polarizabilities of the $\alpha$-carbons directly attached to the carbonyl in **A** was computed and compared, and again in each case the absolute configuration of the major enantiomer of product **C** was unambiguously ascertained by X-ray single crystal structural analysis (using **C** itself or its auxiliary-derived structure **D**, 45 X-ray crystal structures in total). More importantly, as shown in Fig. 1b, direct and quantitative linear free energy mappings between substrates' local polarizabilities and experimentally measured reaction enantioselectivities were afforded in the context of three well-known catalytic systems covering both reductions and oxidations, including Ru-catalyzed Noyori-Ikariya asymmetric transfer hydrogenations of ketones, Corey–Bakshi–Shibata oxazaborolidines-catalyzed reductions of ketones, and Sharpless asymmetric dihydroxylations of

alkenes. We have now uncovered in this work a sensitive dependence of the sense as well as the magnitude of asymmetric induction on the substrate polarizability characteristics. We believe such a long-overlooked electronic effect is a useful addition to our current understanding of the nature of chiral interactions, and thus it could aid rational design and discovery of catalytic enantio-selective processes of higher ees and better efficiency.

## Results and discussions

### Qualitative correlation of local polarizability on enantio-selection

Firstly, to investigate the influences of local polarizability on the sense of enantio-selection, a catalytic system with well-defined mechanism and broad-scope applications should be required. For this purpose, our recently developed Noyori-type Ru-catalyzed asymmetric transfer hydrogenations that are applicable to a wide range of ketone substrates serve as a good platform[17-20]. We computed the local polarizabilities using the D4 dispersion model recently developed by Grimme et al.[21]. In this protocol, the polarizabilities of atoms in different chemical environments are obtained by interpolation and scaling of key parameters such as atomic partial charges and coordination numbers over selected reference molecules. The dynamic polarizabilities of reference atoms and molecules have been pre-determined by time-dependent density functional theory (TD-DFT) methods, which enable the remarkably facile and relatively accurate calculation of local polarizabilities.

For the aryl hetero-aryl ketones, the major enantiomers of the alcohol products and the calculated local $\alpha$-carbon polarizabilities of the ketone substrates are shown in Fig. 2. Although this type of substrates often possesses two structurally as well as sterically comparable substituents, the incorporation of electron-negative heteroatoms is known to lead to the distinctions of their local polarizabilities[22,23]. For instance, the local carbon polarizabilities of phenyl rings are respectively higher than those of 2-pyridyls in ketones **A1** and **A2**, thus leading to the same absolute stereochemistry of the predominant enantiomers for alcohols **C1** and **C2** (Fig. 2a). For aryl and 2-thiazolyl substituted ketones **A3**–**A6**, whether it is phenyl, ortho-substituted

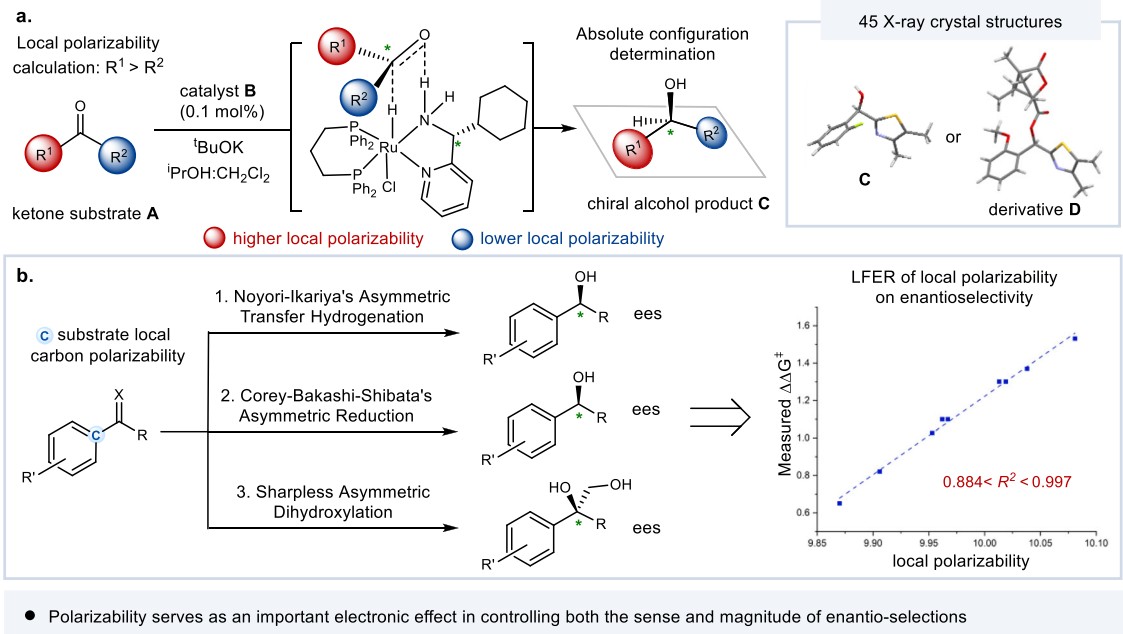

**Fig. 1 | The essential role of local electronic polarizability in enantio-control (this work). a** Correlation of local polarizability on the sense of chiral induction in Ru-catalyzed asymmetric transfer hydrogenation of ketones; **b** Linear free energy relationship (LFER) of substrate local polarizability on enantio-selection in widely known catalytic asymmetric systems. $R^2$: coefficient of determination.

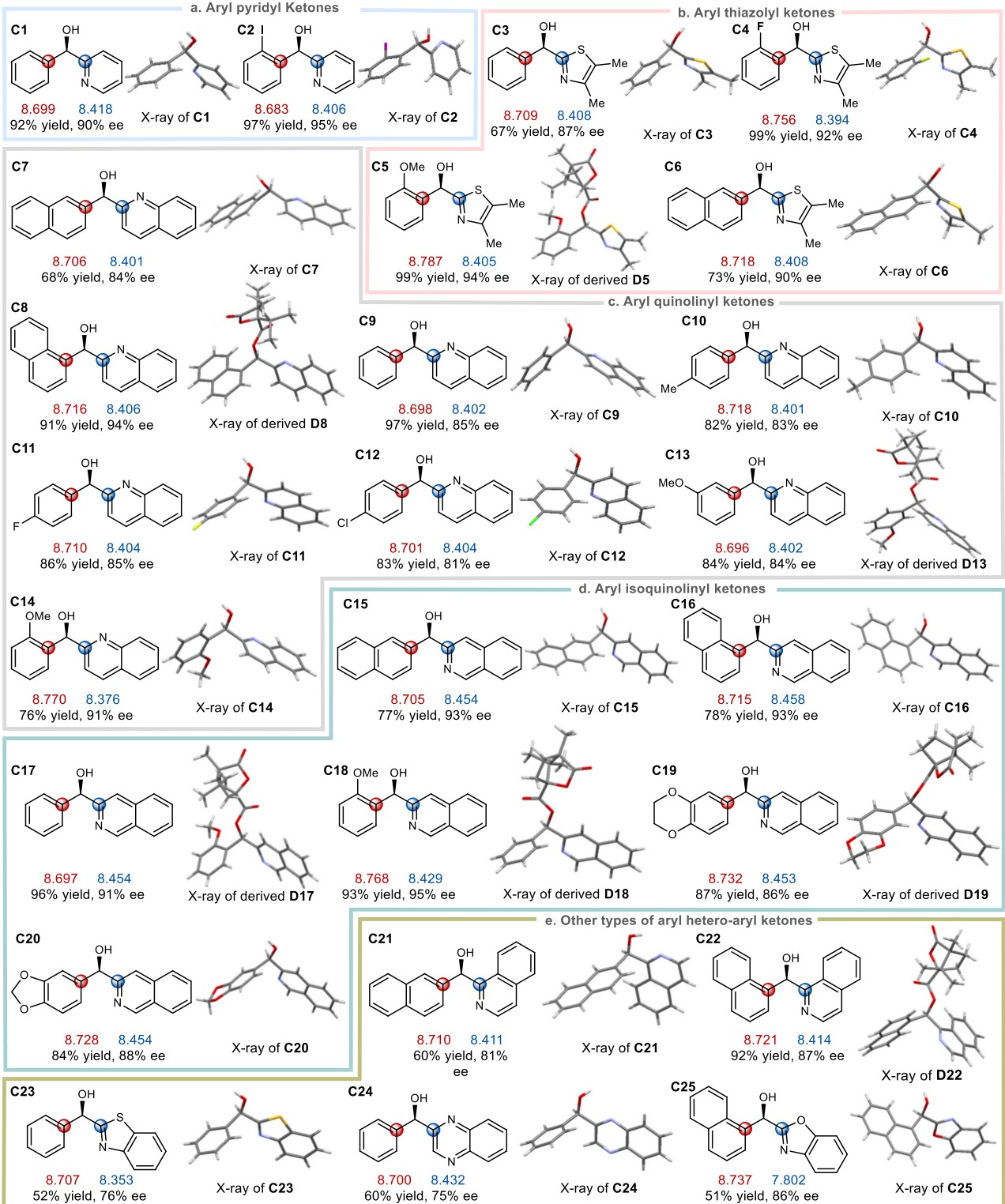

**Fig. 2 | Correlation of substrate substituent local electronic polarizability on the sense of asymmetric induction in Ru-catalyzed asymmetric transfer hydrogenation of aryl hetero-aryl ketones. a** Substrates of aryl pyridyl ketones. **b** Substrates of aryl thiazolyl ketones. **c** Substrates of aryl quinolinyl ketones. **d** Substrates of aryl isoquinolinyl ketones. **e** Other types of aryl hetero-aryl ketones.

The red carbon's local polarizability is larger than the blue carbon's local polarizability. The red and blue numbers underneath each alcohol structures respectively denote the local polarizability values of the red and blue carbon atoms of the corresponding ketones.

phenyl or β-naphthyl, the local carbon polarizabilities of these aryl substituents on the left are higher than those of the 2-thiazolyl rings on the right, therefore the sense of asymmetric induction all remains unchanged (Fig. 2b). For aryl 2-quinolinyl substituted ketones

**A7**–**A14** (Fig. 2c) and aryl 3-isoquinolinyl substituted ketones **A15**–**A20** (Fig. 2d), although different aryl substituents exert more or less influences on their local polarizabilities, in each case the aryl groups on the left predominate in local polarizabilities thus the sense of asymmetric

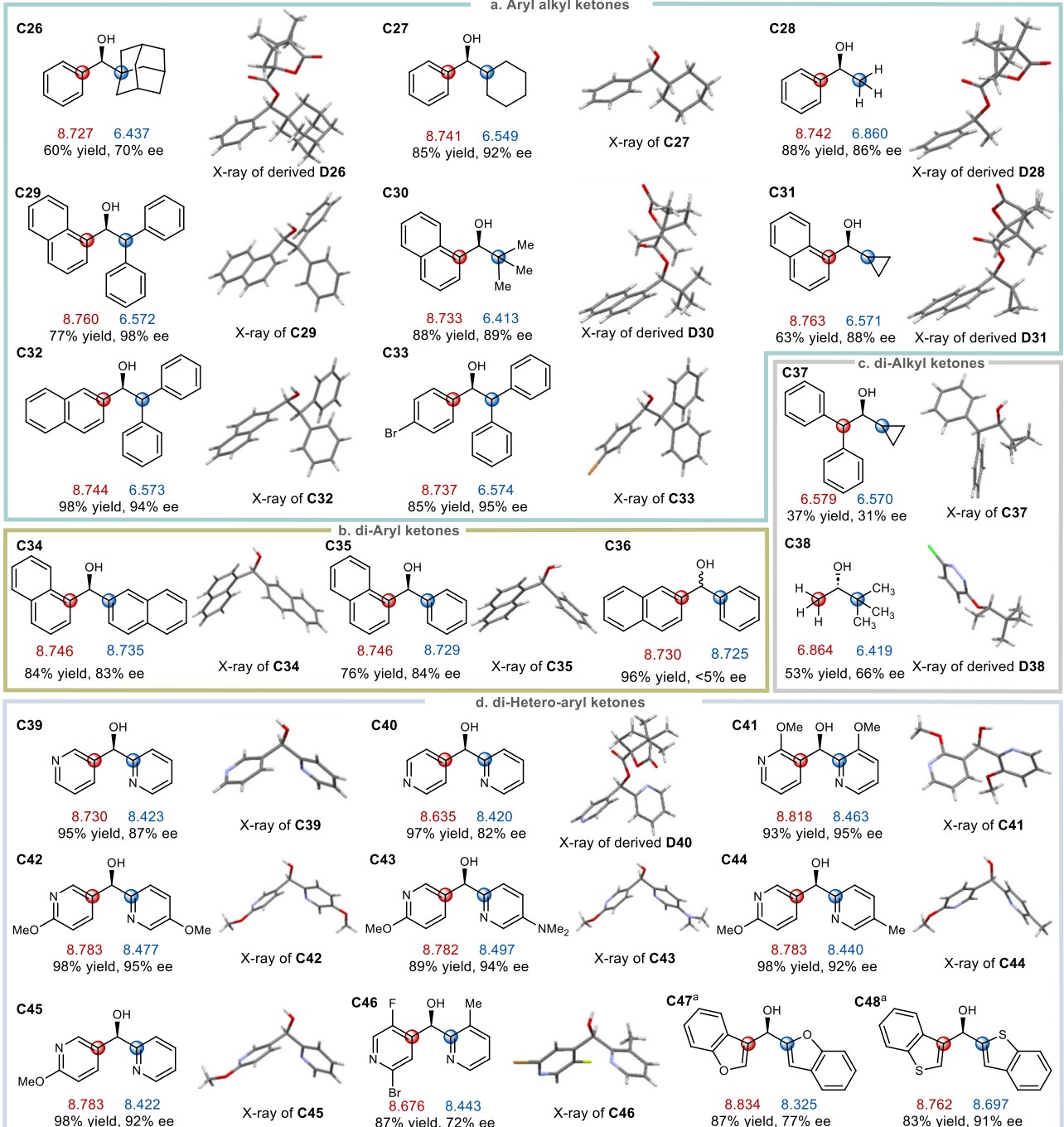

**Fig. 3 | Correlation of substrate substituent local electronic polarizability on the sense of asymmetric induction in Ru-catalyzed asymmetric transfer hydrogenation of aryl alkyl, *di*-aryl, *di*-hetero aryl and *di*-alkyl ketones.** **a** Substrates of aryl alkyl ketones. **b** Substrates of *di*-aryl ketones. **c** Substrates of *di*-alkyl ketones. **d** Substrates of *di*-hetero-aryl ketones. The red carbon's local polarizability is larger than the blue carbon's local polarizability. The red and blue numbers underneath each alcohol structures respectively denote the local polarizability values of the red and blue carbon atoms of the corresponding ketones. The absolute configurations of **C47** and **C48** were determined by ECD (see details in the Supplementary Information).

inductions still remains in consistency. Moreover, some other types of aryl heteroaryl ketones, such as **A21**–**A25**, were also examined (Fig. 2e). As long as the local polarizabilities of aryl groups on the left outweigh those of hetero-aryl substituents on the right, the ketones are all reduced with exactly the same bottom-face selection to furnish their chiral alcohol products **C21**–**C25**.

As shown in Fig. 3, to further demonstrate the influence of local polarizability on the sense of enantio-selection for ketone reductions, other types of substrates, including aryl alkyl ketones **A26**–**A33**, *di*-aryl

ketones **A34**–**A36**, *di*-alkyl ketones **A37**–**A38** and *di*-hetero aryl ketones **A39**–**A48** were next examined. For aryl alkyl ketones (Fig. 3a), the local carbon polarizabilities of aromatic substituents are generally higher than those of aliphatic substituents, thus as long as the aryl group is placed at the left side and the alkyl group is placed at the right side, secondary alcohols **C26**–**C33** with hydroxyl groups facing outward were obtained. A notable comparison exists between ketones **A26, A27** and **A28**: regardless of the steric size differences of the alkyl substituents in these ketones (the adamantyl in **A26**, the cyclohexyl in

**A27**, and the methyl in **A28**), hydrogenations all occurred with the same enantio-facial selection. These results illustrate that there is no doubt that steric effects are important stereochemical elements, however, they could be overridden by competing electronic polarizability effects when a significant substituent polarizability difference is present. For *di*-aryl ketones **A34** and **A35** (Fig. 3b) when substituents with higher local polarizabilities were situated at the left side, the absolute configurations of resulting alcohols in accordance with the above rules were obtained. Although enantio-selection contribution from conducive steric effect cannot be ruled out, the fact that the α-position of naphthyl ring bears higher electronic densities than those of β-position, thereby more labile for accepting electrophiles, echoes well with the selectivity rules widely observed in aromatic electrophilic substitution reactions. An interesting substrate was β-naphthyl phenyl ketone **A36**, which was reduced to alcohol **C36** with almost no ee. This was not unexpected, since β-naphthyl and phenyl substituents have almost the same local polarizabilities according to the calculation (8.730-*versus*−8.725). The cases of *di*-alkyl ketones **A37** and **A38** merit particular attention (Fig. 3c). In the former, the substituent local polarizability difference is exceedingly small (6.579-*versus*−6.570), thus even through the substituent on the left is much bulkier than that on the right, the absence of otherwise synergistic polarizability effect leads to the product **C37** only in a low enantio-selectivity (31% ee). In the latter, the substituent on the left has relatively larger local polarizability but significantly smaller size than that of the substituent on the right. Under such circumstances the weak electronic polarizability influence and pronouncing steric factor are competing against each other and are therefore incapable of delivering the alcohol **C38** in high ee (66% ee recorded) as well as maintaining the same stereochemical integrity (formally an enantio-facial reversal was observed).

Lastly, for *di*-hetero-aryl ketones, in particular a range of pyridyl-pyridyl ketones bearing virtually iso-steric hetero-aromatic rings were examined, the absolute configurations of the resulting alcohols **C39**−**C46** were all in accordance with the polarizability-based stereochemical model (Fig. 3d). For alcohols **C39**−**C42** the two pyridyl ring substituents are essentially identical but just differ in their substitution patterns as well as connection positions to the central carbinols, thereby leading to distinctions in their local carbon polarizabilities and subsequent good-to-high levels of product ees. For alcohols **C43**−**C46** in which each structure has two closely comparable pyridyl rings flanked with different substituents, local polarizability mappings allow for straightforward stereochemical predictions that are in good agreement with experimental observations. For benzofuran or benzothiophene-substituted alcohols **C47**−**C48**, again a simple and subtle switch of ring substitution patterns would enable sufficient local polarizability difference bringing about the observed enantioselections. Overall, these findings disclosed in Figs. 2 and 3 collectively point to the scenario that, although there is no doubt that high enantio-selectivities are reflections of delicate influences from various factors such as electronic, steric, temperature, pressure, solvent, or additive, their respective contributions to enantio-controls are likely not evenly weighted, and polarizability appears to generally serve as a more effective and predominant force.

## Linear free energy relationship studies of substrate local polarizability on enantio-selection

Having established a consistent, polarizability-based enantio-control model for the prediction of major enantiomers formed in the Ru-catalyzed asymmetric transfer hydrogenation of ketones, we would like to further examine the linear free energy relationship (LFER) between substrates' local polarizabilities and observed enantioselectivities in some well-known catalytic asymmetric systems. Within this context the Sigman group has pioneered in the utilization of multivariate linear regression analysis to construct linear free energy relationships in asymmetric catalysis[24,25]. For example, a model connecting substrate and ligand steric effects to enantioselectivity for the propargylation of

aliphatic ketones has been developed[26]. Enantioselectivity has also been correlated to various molecular descriptors in several hydrogenation processes by Wiest, Norrby, and others[27]. Asymmetric catalytic systems that are suitable for this direct and quantitative study should be chemically irreversible, highly reproducible and mechanistically clear. Our Noyori-type Ru-catalyzed asymmetric transfer hydrogenations of ketones with *i*-PrOH as hydrogen source disclosed above are partially reversible[28], thus posing a challenge for such LFER mapping (Upon carefully controlling these reactions at low conversions to prevent the hydrogenation reversibility from eroding the products' enantio-purities, a quantitative LFER between substrates' local polarizabilities and enantioselectivities is revealed here in a series of *di*-aryl ketones where one of the aryl rings are substituted by various electron-donating and withdrawing groups, see Supplementary Fig. 49 in the Supplementary Information). The LFER studies were also conducted by analyzing some exemplary results from both literature-disclosed systems and our own findings where the reactions under concern are irreversible in nature. Illustrated below are three representative catalytic enantioselective processes, covering both chemical reductions and oxidations, whose fine substrate electronic tunings allow for direct visualization of such important implications of electronic polarizability effects on enantio-selections (Fig. 4). These are Ru-catalyzed Noyori-Ikariya asymmetric transfer hydrogenation of ketones with formic acid as hydrogen source (Fig. 4a, from literature results)[29], oxazaborolidines-catalyzed reductions of ketones with the widely known Corey−Bakshi−Shibata catalyst (Fig. 4b, from our own experiments), and OsO$_4$-catalyzed Sharpless asymmetric dihydroxylations of alkenes with a cinchona alkaloid-derived ligand (Fig. 4c, from our own experiments). The local polarizabilities were calculated using a scheme based on the effective volume of atoms in a molecule[30], which was shown to be more accurate than the simple D4-based method in constructing quantitative correlations with free energies. The computational details were compiled in Supplementary Information for reference.

In the Ru-catalyzed transfer hydrogenation of aryl ketones reported independently by Ikariya[29], for the sake of structural comparability, the substrates were categorized into two groups by keeping one of the two aryl substituents constant while systematically varying the other ones bearing different substitutions (Fig. 4a). The substrate local carbon polarizability was tabulated for both of the two groups. In group I (R = *p*-OMe), the decrease in the local polarizabilities of the carbon marked in blue correlated well ($R^2$ = 0.972) with the increase in the difference of free energies of activation ($\Delta\Delta G^{\ddagger}$) derived from experimentally observed reaction enantioselectivities (%ee). In group II (R = H), again good correlation ($R^2$ = 0.938) between local polarizabilities and $\Delta\Delta G^{\ddagger}$ were observed. In the second system, i.e., oxazaborolidines-catalyzed reductions of ketones with the widely-useful Corey−Bakshi−Shibata catalyst, we re-examined the performance of two typical oxazaborolidine catalysts (**D1** and **D2**) on a series of *di*-aryl ketone substrates where the aryl rings are substituted by both electron-donating as well as withdrawing groups. The results were compiled in Fig. 4b, and with such data the corresponding correlation coefficients were readily uncovered ($R^2$ = 0.939 and 0.952 for catalysts **D1** and **D2**, respectively), visualizing excellent linear relationships. In the third system, the classical Sharpless asymmetric dihydroxylations of alkenes by means of a cinchona alkaloid-derived chiral ligand were investigated. In this case the commercially available *bis*-cinchona alkaloids such as AD-mix ligands were not suitable as they both generally lead to closely comparably high enantioselectivities over a broad range of substrates[31]. To this aim, a series of *mono*-cinchona alkaloid ligands that normally lead to moderate to good enantioselectivities were synthesized and examined under otherwise identical Sharpless asymmetric dihydroxylation conditions[31], and ligand **E** was found to be the optimal ligand for LFER studies (See Supplementary Information for more details). As shown in Fig. 4c, the results on a series of virtually *iso*-steric alkene substrates were

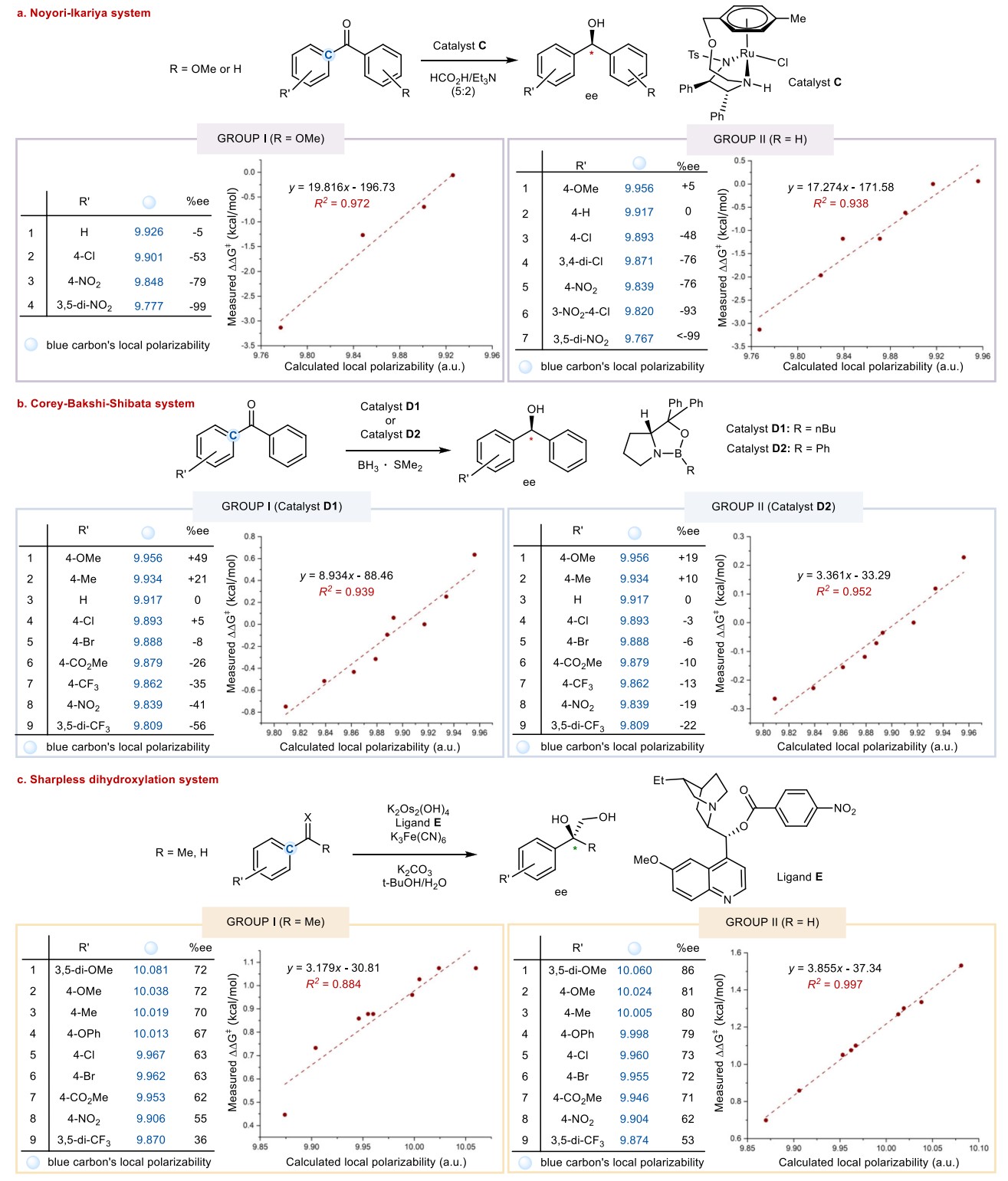

**Fig. 4 | Linear free energy relationship (LFER) analysis of substrate local polarizabilities on enantio-selections. a** Asymmetric transfer hydrogenation of *di*-aryl ketones by means of a Noyori-Ikariya catalyst. **b** Asymmetric reduction of *di*-aryl ketones catalyzed by two Corey−Bakshi−Shibata oxazaborolidine catalysts. **c** Sharpless asymmetric dihydroxylation of aryl substituted terminal alkenes by a cinchona alkaloid-derived ligand. Atomic units (a. u.).

found to be electronically well differentiated and conducive for constructing linear correlations. For the two sets of terminal alkenes, both 1,1-*di*-substituted (group I, R = Me) and mono-substituted (group II, R = H) structural modes, good to excellent linear correlations between local polarizabilities and reaction $\Delta\Delta G^{\ddagger}$ were deciphered (for group I: $R^2 = 0.884$; for group II: $R^2 = 0.997$). Collectively, these findings

demonstrated herein that, by choosing reaction systems with minimal steric effect-derived interference, direct and quantitative linear free energy correlations between the substrates' local polarizabilities and the observed magnitudes of reaction enantio-selection could be consistently identified, and they help shed light on the important role of polarizability electronic effect on the origin of enantio-selection.

In conclusion, we have described herein, with combined experimental and computational investigation, the essential role of substrate local polarizabilities in predicting the sense and magnitude of asymmetric induction. To this aim a wide range of ketone substrates, including aryl hetero-aryl, aryl alkyl, *di*-aryl, *di*-hetero-aryl and *di*-alkyl ketones, were prepared and used to investigate the role of carbonyl substituents' local polarizabilities in determining the sense of asymmetric induction in the Ru-catalyzed Noyori transfer hydrogenation. In each case, the local polarizabilities of the α-carbons directly attached to the central carbonyl group were computed and compared, and the absolute stereochemical configuration of the alcohol product was unambiguously ascertained by means of *X*-ray crystallography. Furthermore, by employing experimental data from both literature disclosures and our own studies, direct and quantitative linear free energy correlations between the substrate local polarizabilities and the observed magnitudes of reaction enantio-selection ($\Delta\Delta G^{\ddagger}$ values) were consistently uncovered in three illustrative catalytic enantioselective processes, i.e., reductions of aryl ketones by Noyori-Ikariya catalyst as well as Corey−Bakshi−Shibata protocol, and oxidation of alkenes with Sharpless asymmetric dihydroxylation. Given the considerable advances achieved in the accurate and efficient computation of local polarizabilities, the development and implementation of such a strategy should be feasible. It is thus envisioned that a judicious combination of polarizability as well as steric effects, when considered on both substrate and catalyst aspects, would enable rational design of future catalytic enantio-selective processes of higher efficiency for better production of useful chiral substances.

## Methods

### General procedure for asymmetric transfer hydrogenation of ketones A and determination of the absolute stereochemistry of the reduced alcohols C

In an argon filled-glovebox, catalyst **B** (0.2–2.0 µmol, 0.1–1.0 mol%) was suspended in 2-propanol (2.0 mL) in a 4-mL vial, and potassium *tert*-butanolate (30 µL, 1 M in *tert*-butanol, 15 mol%) was added. After the mixture was stirred for 20 min, ketone **A** (0.2 mmol) dissolved in $CH_2Cl_2$ (1.0 mL) was added. After stirring at room temperature for 2 h, the reaction mixture was concentrated under reduced pressure and the crude product was purified by silica gel column chromatography using hexane/EtOAc to give the alcohol **C**. In order to determine the absolute configuration of the major enantiomer of product **C**, its minor enantiomer was removed by preparative HPLC (OD-H or AD-H 2.0*25 cm, 5 µm). Finally, the *X*-ray crystal of the predominant enantiomer of **C** (100% ee) or its chiral auxiliary-derived **D** was obtained by liquid/liquid diffusion with $CH_2Cl_2$/hexane or THF/hexane.

### Procedure of calculation of local polarizability

Initial molecular geometries were optimized with *Gaussian 16* using the B3LYP functional. The SDD effective core potential (ECP) was chosen to describe iodine, and for other atoms the 6-311 + G(d,p) basis set was used. Frequency calculations were carried out at the same level of theory to identify all of the stationary points as minima (zero imaginary frequency). For each molecule, its conformation having the lowest free energy is selected to calculated the corresponding atomic polarizabilities based on Grimme's D4 dispersion model using the GFN2-xTB 6.3 program.

## Data availability

All data generated or analyzed during this study are available within the paper and its supplementary information files. Crystallographic data for the structures reported in this article have been deposited at the Cambridge Crystallographic Data Center, under deposition numbers CCDC 2017192 (**C1**), CCDC 1897334 (**C2**), CCDC 2017080 (**C3**), CCDC 2017098 (**C4**), CCDC 2017150 (**D5**), CCDC 2017187 (**C6**), CCDC 2017094 (**C7**), CCDC 2017193 (**D8**), CCDC 2017022 (**C9**), CCDC 2017093 (**C10**), CCDC 2017091 (**C11**), CCDC 2017092 (**C12**), CCDC 2017194 (**D13**), CCDC 2017096 (**C14**), CCDC 2017134 (**C15**), CCDC 2017106 (**C16**), CCDC 2017196 (**D17**), CCDC 2017157 (**D18**), CCDC 2017161 (**D19**), CCDC 2017154 (**C20**), CCDC 2017185 (**C21**), CCDC 2183982 (**D22**), CCDC 2017100 (**C23**), CCDC 2017097 (**C24**), CCDC 2017199 (**C25**), CCDC 2017197 (**D26**), CCDC 2183980 (**C27**), CCDC 2255377 (**D28**), CCDC 2017205 (**C29**), CCDC 2017173 (**D30**), CCDC 2018330 (**D31**), CCDC 2017130 (**C32**), CCDC 2017175 (**C33**), CCDC 2017293 (**C34**), CCDC 2017162 (**C35**), CCDC 2181768 (**C37**), CCDC 2183983 (**D38**), CCDC 2017188 (**C39**), CCDC 2208981(**D40**), CCDC 2019561 (**C41**), CCDC 2311153 (**C42**), CCDC 2311152 (**C43**), CCDC 2311154 (**C44**), CCDC 2311155 (**C45**), CCDC 2017295 (**C46**), CCDC 2235591(**C74**), CCDC 2017208 (Ru-catalyst **B**). Copies of the data can be obtained free of charge via https://www.ccdc.cam.ac.uk/structures/. Source data are provided with this paper.

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

## Acknowledgements

We are grateful for financial support from the Shenzhen Nobel Prize Scientists Laboratory Project (C17783101 to X.X.), Guangdong Provincial Key Laboratory of Catalysis (2020B121201002), Guangdong Basic and Applied Basic Research Foundation (2021A1515010329 to X.X.), the National Natural Science Foundation of China (22171130 to P.Y.) and Shenzhen Youwei Tech Group. Computational work was supported by Center for Computational Science and Engineering at Southern University of Science and Technology (SUSTech) and the CHEM high-performance supercomputer cluster (CHEMHPC) at Department of Chemistry, SUSTech.

## Author contributions

F.C., D.Z.W. and X.X. developed and conducted the reactions. Y.C. and P.Y. conducted the computational studies. X.X., P.Y. and D.Z.W. designed and directed the investigation and prepared the manuscript. All correspondence should be sent to X.X. (xingxy@susteche.edu.cn), P.Y. (yupy@sustech.edu.cn) and C.X. (xuc@sustech.edu.cn).

## Competing interests

The authors declare no competing interests.
