## [Peer Review File · Nature Communications]

Reviewers' Comments:

Reviewer #1:

Remarks to the Author:

In this manuscript, Chen et. al. disclose the usage of local polarizability as a predictive model for the absolute sense and, in some cases, magnitude of enantioselectivity. Two different types of data are provided to support this claim. In figures 2 and 3, the authors examine a Ru-catalyzed transfer hydrogenation of ketones they have reported previously. For this reaction, the authors examine the local polarizability of the two α -carbons and use this to predict the absolute sense of enantioinduction. An advantage of this system is that it tolerates a wide range of substituents, including heteroarenes and alkyl groups. The possibility for local polarizability to provide a general model across a wide range of different classes of substituents is the most interesting aspect of this paper, so the examination of such different substrates is crucial. However, the authors note that this reaction may be partially reversible, potentially ablating the e.e. of the product, and thus, precluding the development of a quantitative model. It would have been far more interesting if the authors could have shown a quantitative relationship between e.e. and local polarizability in this system (or any system tolerating a wide range of different substituents), and I question whether they could have just run this reaction to low conversions to minimize racemization of the product. In figure 4, the authors present quantitative correlations between local polarizability and enantioselectivity in three other reactions: a different Ru-catalyzed transfer hydrogenation, the CBS reduction, and the SAD. Unfortunately, the range of substrates examined in these reactions is much narrower, with the primary variation being probed consisting of simple variations in meta- and para-substituents on a phenyl ring. Given the wide range of existing strategies for correlating the properties of arenes with reaction outcomes such as the classic Hammett LFER and more modern studies from computational groups such as Wheeler, there is a very high bar for a new LFER being useful if it is restricted to substituted phenyl rings. Overall, I feel that this work is potentially interesting in that local polarizability may be a useful model for predicting the sense and magnitude of enantioinduction for some classes of reactions, but the current data does not clearly demonstrate the superiority of this approach over existing methods. For consideration in a journal of the quality of Nature Communications, the authors will need to collect substantial amounts of additional data probing edge cases where other models such as visual inspection of the substrates or Hammett plots fail. There are also significant problems with the presentation of both precedent and this work that need to be corrected.

In the abstract and introduction the authors state that "The origin of enantio-selection in chiral induction events is usually thought to be consequences of steric, electronic, or conformational effects... the underlying connection between polarizability and enantio-selection has long been largely overlooked," and "conventional wisdom usually attributes the origin of enantio-selection to consequences of steric or stereo-electronic effects." I do not feel that this description is an accurate assessment of the current state of thinking about enantioinduction within the field of asymmetric catalysis. While it is true that most early models of enantioinduction were based largely on sterics, it has long been understood that a wide range of attractive interactions—including interactions such as dispersion forces and cation- π interactions, the strength of which depends at least in part on polarizability—can contribute to or even be dominant factors in asymmetric induction. One notable early example is the proposal from Sharpless that a π -stacking interaction contributes to enantioinduction in the asymmetric OsO₄-catalyzed dihydroxylation, a particularly apt example since this is one of the reactions the authors study in this manuscript. Given that many prominent research programs (Jacobsen, Toste, Miller, Phipps, etc.) are based at least in part on these strategies and that leading scholars of computational chemistry (Houk, Patton, Wheeler, Liu, etc.) commonly propose dispersive interactions as being at least partially responsible for enantioinduction, the statements made in the introduction and abstract need to be significantly modified. These sections should be re-written to focus on what is novel in this paper, which is the focus on local polarizability at a single site, rather than polarizability in general and non-steric models of enantioinduction.

The statement that the successes of enantioselective catalysis "have been almost entirely concentrated on work involving [aryl-alkyl] substrate" and that "for di-alkyl substituted substrates 9, despite years of efforts, there have rarely been any reports with high ees," are also inaccurate. It is certainly true that these substrate classes are more difficult and are less commonly reported, but there are still numerous examples of reactions that give high e.e. in substrates lacking arenes. The big picture claim—that these substrates are more difficult and less commonly successful—can

be retained, but it should be toned down.

A deeper concern is the fact that, at least in some places, the manuscript presents local polarizability as being responsible for enantioinduction (e.g., statements such as "it appears to be more effective factor governing both of the sense and magnitude of enantio-selection" and "we have described herein, with combined experimental and computational investigation, the essential role of substrate local polarizabilities in determining the sense and magnitude of asymmetric induction"). Unfortunately, no data is provided in this paper to support a causative rather than a correlative relationship. In fact, the discussion in Figs. S49 and S50 seem to explicitly acknowledge the correlative nature of their model. All discussions of local polarizability controlling (rather than predicting) enantioinduction need to be removed, or the authors need to provide significant additional data supporting a causal relationship between local polarizability and enantioinduction. The weaker claim—that local polarizability correlates with and can be predictive of the absolute sense and magnitude of enantioinduction—is better supported by the data in the paper, but some aspects of this are problematic. With regard to the transfer hydrogenation presented in Figs. 2 and 3, very few of the substrates provide a firm test of the hypothesis. In Figure 2, every substrate contains a simple arene on the left hand side (C19 and C20 do have heterocycles on the left hand side, but the ring attached to the carbonyl is a simple arene) and heterocycles on the right hand side. While it is true that the sense of enantioinduction can be predicted based on local polarizability, it can also be predicted based on visual inspection of the substrates to see which ring is a heterocycle. Likewise, the sense of enantioinduction for C26-C33 in Figure 3 can be predicted trivially based on which substituent is alkyl. Compounds C34-C42 are more interesting, but only offer mixed support for the authors' proposal. I strongly disagree with the authors statement that "the α -versus- β local carbon polarizabilities distinction is sufficient for bringing about the recorded high ees in C34." It is extremely common for ortho substitution, and especially, for ortho- \rightarrow -ring fusions to have a pronounced effect on enantioselectivity, which is typically attributed to changes in the shape of the substrate or partial disruption of conjugation due to a change in the dihedral angle between the arene and the substituent. Moreover, the differences in local polarizability for these compounds (0.011 for C34, 0.017 for C35) are very small compared to those in other highly performing substrates (>0.3 for all highly performing substrates and typically >1), and instead, are much more similar to poorly performing compounds like C36 (0.005), which the authors state was expected to give low e.e. due to the similarity in the local polarizability of the two substituents, and in C41 (0.009), which was formed in low e.e. To me, compounds C34 and C35 seem to be an obvious example of local polarizability failing to accurately predict enantioselectivity, as they seem to predict extremely low e.e. However, if the authors truly believe that very small differences in local polarizability (differences <0.1) then they should demonstrate this experimentally. For example, ketones containing two simple arenes with different p-substituents are likely to have local polarizability differences on that order of magnitude. Does the model correctly predict e.e. for a compound like phenyl-p-tolyl ketone? Likewise, in C42 the model predicts the wrong enantiomer despite a large difference in local polarizability (0.445 – a difference larger than those in most of C1-C24 and C37-C40). The authors attribute this to the steric hindrance of the t-butyl group, which is reasonable, but then why were less hindered methyl alkyl ketones (e.g., methyl-cyclohexyl ketone or methyl cyclopropyl ketone, which would still have a large difference in local polarizability of the two substituents) not tested? Compounds C37-C40 are by far the strongest and most interesting evidence in support of the central argument of this paper as, unlike the diaryl ketones in Fig. 2, they cannot be predicted based on trivial visual inspection. However, these 4 examples are not sufficient evidence. For the argument in Figs. 2 and 3 to be compelling, the authors need to provide more examples where alternate, trivial methods of prediction will fail. I would be particularly interested to see examples of ketones where one side is a simple arene and the other is a heterocycle with a local polarizability exceeding that of the arene (so the predicted sense of enantioinduction would be opposite that in Fig. 2 if the compounds are drawn in the same orientation). I also wonder what happens with cyclohexenes (if they are tolerated by the reaction. As mentioned above, it would also be far more compelling if the authors could find a way of demonstrating a quantitative relationship (as they shown in Figure 4) across substrates with widely varied substituents (as they show in Figure 2 and 3).

In figure 4, the authors correlate local polarizability with e.e. in 3 different reactions. In contrast to the data in Figs. 2 and 3, the compounds used for Fig. 4 are all simple arenes with varied substituents, making them amenable to classic LFERs such as Hammett plots. The authors actually show Hammett plots for these data in Fig. S48, with the result that correlations comparable to those shown in Fig. 4 are obtained via a straightforward Hammett analysis for all but the Sharpless

dihydroxylation (and even then, it seems that only a single substituent, 3,5-dimethoxy benzene, was predicted inaccurately). Given that the Hammett analysis is much easier to perform than computational analysis of local polarizability, it is not clear to me that the value of predicting that one substrate more accurately is worth the added complications of using their approach. I also wonder how the correlation with local polarizability compares to measurement of the polarizability of the entire aryl ring, or other properties that can be generated computationally such as the electrostatic potential at a fixed distance above the ring.

There may be significant advantages to using local polarizability over alternatives (visual inspection, Hammett analysis, other computational approaches) in terms of the ease of the analysis (where visual inspection and Hammett analysis is easier, but other computational approaches may be more challenging), the accuracy of the predictions, or the generality of the approach (this strikes me as the most likely advantage of this approach as Hammett analyses can only easily be performed for meta- and para-substituted phenyl rings), but it is not possible to reach a conclusion either way from the data presented here. The manuscript does provide a handful of examples of this: compounds C37-C40 and the 3,5-dimethoxystyrene in the SAD, but the vast majority of the examples in the current paper could be more easily predicted by other means. I would be happy to reconsider a revised version of this manuscript where the authors have addressed these concerns and provide stronger evidence for the utility of models based on local polarizability over other alternatives.

Reviewer #2:

Remarks to the Author:

Key Results

The authors report that local polarizability of atoms vicinal to a substrate's prochiral center predicts the sense and magnitude of enantioinduction. Remarkably, this trend is recapitulated across diverse catalysts, mechanisms, and transformations. These results provide a compelling model for interpreting low enantioselectivities observed in asymmetric hydrogenation of di-alkyl ketones, especially for substrates with large differences in the steric profiles of the two substituents. The authors make clear that polarizability is the most important, but not the only factor that contributes to enantioinduction (steric effects play a role). The authors note that although the Hammett parameter σ is correlated with enantioselectivity, the free energy of enantioinduction is more correlated with polarizability than σ .

Validity

The results in this paper show a clear correlation between polarizability and enantioinduction. However, this work also shows a clear correlation between σ and enantioinduction. In the absence of the polarizability data, the title of this paper could have easily been "Sigma matters for enantioinduction". This motivates the question, what other factors that have not been considered might account for enantioinduction? Some obvious candidates include factors that are correlated with local (atom-wise) polarizability, namely attributes of the fragment (the whole substituent attached to the pro-chiral center) including polarizability, quadrupole moment, and dispersion interaction energy. Note that the title of this paper could easily be "Dispersion matters for enantioinduction" and this would fall in line with the work of others (see Schreiner reference below).

Since local polarizability cannot be experimentally measured, we cannot determine their error. Without knowing how much error is associated with the measurement, it is perilous to draw conclusions based on the magnitude of the correlation.

Significance

This study has the potential to be highly significant, but that potential is not fully clear in the current manuscript. As it stands, the paper provides an impressive collection of correlations that span diverse and important transformations. It appears that there is an important truth lying at the heart of these data. However, there is no piercing insight drawn from these correlations.

There are several reports that note correlations between free-energy of enantioinduction and polarizability (or polarizability per volume). It is concerning that none of these reports are cited in the present manuscript (especially the report of Schreiner, which closely aligns with the results reported here on the Corey-Bakshi-Shibata reduction).

<https://onlinelibrary.wiley.com/doi/full/10.1002/anie.202012760> (figure 12)

<https://pubs.acs.org/doi/10.1021/ja101256v> (discussed in main text, figure in SI, pg S-34)

<https://www.nature.com/articles/nchem.1450> (discussed in main text, figure in SI, pg S-27)

Data and methodology

The data and methodology are sound.

Analytical approach

Noting the R² value for two independent variables (sigma vs local polarizability) is not exactly robust, but it is not far from the standards of the field.

Improvements

This work starts with the problem of challenging substrate classes. It provides a compelling model for why those substrates are challenging, but offers no solutions. Since this work does not offer a solution to achieving high enantioselectivity with this class of substrates, it might be prudent to omit them from the introductory figure.

Clarity and context

The authors do a great job of introducing an important problem and clearly presenting compelling data that relates to the problem. There is a lack of clarity in the interpretation of the data. It seems that the authors are essentially noting a correlation and then avoiding any interpretation of the correlation. If this is the case, they should state explicitly that the physical basis of the correlation and the extent to which it applies to other asymmetric catalytic reactions is outside the scope of this work. As it is currently written, the manuscript indirectly suggests that local polarizability is the dominant driver of enantioselectivity in all of asymmetric catalysis (see, for example, the first sentence of the conclusion).

References

The Schreiner paper above should be cited. It would be appropriate to cite other works (like those noted above) that have noted a correlation between enantioselectivity and polarizability.

Reviewer #3:

Remarks to the Author:

The work is interesting by novelty is lacking the catalyst itself is a chiral molecule, therefore can lead to the enantioselective catalysis. Polarizability in this case is not helping to achieve enantiopurity, the results are highly speculative.

XRD figure have problems in the descriptions, you show them in "Sticks" mode why then are you taking about 50% probability in ellipsoids if you do not show those figures in an ellipsoids mode? What a nonsense!

Reviewer #4:

Remarks to the Author:

Fumin Chen et al. report an experimental and theoretical study of three enantioselective reactions and suggest that local polarizability differences between two α -Cs of substituents attached to a carbonyl correlate with enantioselectivity. In particular, large and small are not effective, while alkyl (not very polarizable) and aryl (polarizable) are effective in promoting enantioselectivity. The examples all involve aryl alkyl ketones (with very few vs. dialkyl or diaryl for comparison).

The authors report experimental data on Ru-catalyzed asymmetric transfer hydrogenation of unsymmetrical aryl-aryl ketones. The e.e.s. are mostly 75-95%. The aryl-alkyl ketones in Figure 3 are also about the same ee range except for C41 and C42, dialkyl alkyl, which are 31% and 66%. For three reactions, Figure 4 shows good correlation between measured $\Delta\Delta G^\ddagger$ and the local polarizability difference of substituents. No error bars are given, but R² values are given.

They do the same for two other reactions.

The authors describe the calculations of local polarizabilities "using the D4 dispersion model" or Grimme. While a few words about how this is done are given, actually the reader does not know how this was done even after attempting to read Grimme's 19-page paper.

Although the SI notes a little more detail, use of MO6-2X but then GFN2-xTB, a simple semi-empirical tight-binding method, it will still not be possible to reproduce the polarizabilities in this paper without trial and error testing. Another method with Multiwfn was described also.

The paper is impressive in the large amount of data – mostly presented in the SI – and correlations. However, there is no attempt to explain why this works. An energy decomposition analysis of a few cases might help – do the stabilizations resulting from polarization parallel polarizabilities?

In summary, the paper is impressive in (1) the amount of data generated almost without discussion and (2) the discovery that the local polarizability of α -Cs correlates with selectivity. But we do not really know what is meant by local polarizability or why it correlates.

Reviewer 1:

In this manuscript, Chen et. al. disclose the usage of local polarizability as a predictive model for the absolute sense and, in some cases, magnitude of enantioselectivity. Two different types of data are provided to support this claim. In figures 2 and 3, the authors examine a Ru-catalyzed transfer hydrogenation of ketones they have reported previously. For this reaction, the authors examine the local polarizability of the two α -carbons and use this to predict the absolute sense of enantioinduction. An advantage of this system is that it tolerates a wide range of substituents, including heteroarenes and alkyl groups. The possibility for local polarizability to provide a general model across a wide range of different classes of substituents is the most interesting aspect of this paper, so the examination of such different substrates is crucial. However, the authors note that this reaction may be partially reversible, potentially ablating the e.e. of the product, and thus, precluding the development of a quantitative model. It would have been far more interesting if the authors could have shown a quantitative relationship between e.e. and local polarizability in this system (or any system tolerating a wide range of different substituents), and I question whether they could have just run this reaction to low conversions to minimize racemization of the product.

Our response: We thank this reviewer for the constructive comments and valuable suggestions. We agree with the reviewer that it would be really exciting to demonstrate a quantitative correlation for such a diverse substitution pattern. Unfortunately, it is not possible as both of the ketone substituents are simultaneously varied. Alternatively, in order to show a quantitative relationship between e.e. and local polarizability in our Ru-catalyzed transfer hydrogenation system, we further examined a series of *di*-aryl ketones whose reversible dehydrogenations are relatively slow. Furthermore, we examined the ees of products at low conversion within only 5 min. By varying the substituents on one of the aryl rings, the measured $\Delta\Delta G^\ddagger$ and computed local polarizability at the alpha carbon demonstrated good correlation for this system. The results are shown in the following scheme, which has been added as Figure S49 in the revised Supporting Information. The corresponding tests have also been added into the revised manuscript which were highlighted in yellow: “Asymmetric catalytic systems that are suitable for this direct and quantitative study should be chemically irreversible, highly reproducible and mechanistically clear. Our Noyori-type Ru-catalyzed asymmetric transfer hydrogenations of ketones with *i*-PrOH as hydrogen source disclosed above are partially reversible²⁸, thus posing a challenge for such LFER mapping (Upon carefully controlling these reactions at low conversions to prevent the hydrogenation reversibility from eroding the products’ enantio-purities, a quantitative LFER between substrates’ local polarizabilities and enantioselectivities is revealed here in a series of *di*-aryl ketones where one of the aryl rings are substituted by various electron-donating and withdrawing groups, see Fig. S49 in the Supplementary Information).”

Figure S49 in the revised manuscript

In figure 4, the authors present quantitative correlations between local polarizability and enantioselectivity in three other reactions: a different Ru-catalyzed transfer hydrogenation, the CBS reduction, and the SAD. Unfortunately, the range of substrates examined in these reactions is much narrower, with the primary variation being probed consisting of simple variations in meta- and para-substituents on a phenyl ring. Given the wide range of existing strategies for correlating the properties of arenes with reaction outcomes such as the classic Hammett LFER and more modern studies from computational groups such as Wheeler, there is a very high bar for a new LFER being useful if it is restricted to substituted phenyl rings.

Overall, I feel that this work is potentially interesting in that local polarizability may be a useful model for predicting the sense and magnitude of enantioinduction for some classes of reactions, but the current data does not clearly demonstrate the superiority of this approach over existing methods. For consideration in a journal of the quality of Nature Communications, the authors will need to collect substantial amounts of additional data probing edge cases where other models such as visual inspection of the substrates or Hammett plots fail. There are also significant problems with the presentation of both precedent and this work that need to be corrected.

Our response: We really appreciate this reviewer's valuable suggestions and comments. Additional experimental work has been conducted to address these concerns, in particular on some structures which we believe represent the "edge cases". Furthermore, the recognition of literature precedents as well as presentation of current work has been substantially re-written. The issue was tackled in a low-key manner. Please see the details in the following.

1. In the abstract and introduction, the authors state that "The origin of enantio-

selection in chiral induction events is usually thought to be consequences of steric, electronic, or conformational effects... the underlying connection between polarizability and enantio-selection has long been largely overlooked,” and “conventional wisdom usually attributes the origin of enantio-selection to consequences of steric or stereo-electronic effects.” I do not feel that this description is an accurate assessment of the current state of thinking about enantioinduction within the field of asymmetric catalysis. While it is true that most early models of enantioinduction were based largely on sterics, it has long been understood that a wide range of attractive interactions—including interactions such as dispersion forces and cation- π interactions, the strength of which depends at least in part on polarizability—can contribute to or even be dominant factors in asymmetric induction. One notable early example is the proposal from Sharpless that a π stacking interaction contributes to enantioinduction in the asymmetric OsO₄-catalyzed dihydroxylation, a particularly apt example since this is one of the reactions the authors study in this manuscript. Given that many prominent research programs (Jacobsen, Toste, Miller, Phipps, etc.) are based at least in part on these strategies and that leading scholars of computational chemistry (Houk, Patton, Wheeler, Liu, etc.) commonly propose dispersive interactions as being at least partially responsible for enantioinduction, the statements made in the introduction and abstract need to be significantly modified. These sections should be re-written to focus on what is novel in this paper, which is the focus on local polarizability at a single site, rather than polarizability in general and non-steric models of enantioinduction.

Our response: according to this reviewer’s valuable suggestions, the Abstract and Introduction sections have been re-written. The discussions of non-steric models of enantioinduction have been removed, and the focus is placed on polarizability effect alone, on both of its qualitative correlation on sense of enantio-selection and quantitative linear free energy relationship (LFER) analysis. Please see the details in the revised manuscript.

2. The statement that the successes of enantioselective catalysis “have been almost entirely concentrated on work involving [aryl-alkyl] substrate” and that “for di-alkyl substituted substrates⁹, despite years of efforts, there have rarely been any reports with high ees,” are also inaccurate. It is certainly true that these substrate classes are more difficult and are less commonly reported, but there are still numerous examples of reactions that give high e.e. in substrates lacking arenes. The big picture claim—that these substrates are more difficult and less commonly successful—can be retained, but it should be toned down.

Our response: in the revised manuscript, the introduction has been re-written, the factual experimental work was enhanced, and the corresponding claim had been toned down. Furthermore, the description on di-alkyl substrates has been removed.

3. A deeper concern is the fact that, at least in some places, the manuscript presents local polarizability as being responsible for enantioinduction (e.g., statements such as “it appears to be more effective factor governing both of the sense and magnitude of enantio-selection” and “we have described herein, with combined experimental and computational investigation, the essential role of substrate local polarizabilities in determining the sense and magnitude of asymmetric induction”). Unfortunately, no data is provided in this paper to support a causative rather than a correlative relationship. In fact, the discussion in Figs. S49 and S50 seem to explicitly acknowledge the correlative nature of their model. All discussions of local polarizability controlling (rather than predicting) enantioinduction need to be removed, or the authors need to provide significant additional data supporting a causal relationship between local polarizability and enantioinduction.

Our response: According to this reviewer’s suggestion, statements including “controlling, governing or determining, etc in describing the relationship between local polarizability and enantioselections” have been toned down and revised as “predicting”. Please see the details in the revised manuscript.

4. The weaker claim—that local polarizability correlates with and can be predictive of the absolute sense and magnitude of enantioinduction—is better supported by the data in the paper, but some aspects of this are problematic. With regard to the transfer hydrogenation presented in Figs. 2 and 3, very few of the substrates provide a firm test of the hypothesis. In Figure 2, every substrate contains a simple arene on the left hand side (C19 and C20 do have heterocycles on the left hand side, but the ring attached to the carbonyl is a simple arene) and heterocycles on the right hand side. While it is true that the sense of enantioinduction can be predicted based on local polarizability, it can also be predicted based on visual inspection of the substrates to see which ring is a heterocycle. Likewise, the sense of enantioinduction for C26-C33 in Figure 3 can be predicted trivially based on which substituent is alkyl.

Compounds C34-C42 are more interesting, but only offer mixed support for the authors’ proposal. I strongly disagree with the authors statement that “the α -versus- β local carbon polarizabilities distinction is sufficient for bringing about the recorded high ees in C34.” It is extremely common for ortho substitution, and especially, for ortho-ring fusions to have a pronounced effect on enantioselectivity, which is typically attributed to changes in the shape of the substrate or partial disruption of conjugation due to a change in the dihedral angle between the arene and the substituent. Moreover, the differences in local polarizability for these compounds (0.011 for C34, 0.017 for C35) are very small compared to those in other highly performing substrates (>0.3 for all highly performing substrates and typically >1), and instead, are much more similar to poorly performing compounds like C36 (0.005), which the authors state was expected to give low e.e. due to the similarity in the local polarizability of the two substituents, and in C41 (0.009), which was formed in low e.e. To me, compounds C34 and C35 seem to be an

obvious example of local polarizability failing to accurately predict enantioselectivity, as they seem to predict extremely low e.e. However, if the authors truly believe that very small differences in local polarizability (differences <0.1) then they should demonstrate this experimentally. For example, ketones containing two simple arenes with different para-substituents are likely to have local polarizability differences on that order of magnitude. Does the model correctly predict e.e. for a compound like phenyl-p-tolyl ketone?

Likewise, in C42 the model predicts the wrong enantiomer despite a large difference in local polarizability (0.445 – a difference larger than those in most of C1-C24 and C37-C40). The authors attribute this to the steric hindrance of the t-butyl group, which is reasonable, but then why were less hindered methyl alkyl ketones (e.g., methyl-cyclohexyl ketone or methyl cyclopropyl ketone, which would still have a large difference in local polarizability of the two substituents) not tested?

Our response: we appreciate this reviewer's thoughtful suggestions and comments. In Figures 2 and 3, we aim to demonstrate the power of local polarizability in predicting the major enantiomer formed in the Ru-catalyzed ATH of ketones. Thus, firstly, the statement "the α -versus- β local carbon polarizabilities distinction is sufficient for bringing about the recorded high ees in C34" more implied a quantitative relationship between local polarizability and ees, which was not supported by the results here. This statement indeed causes confusion and has been deleted in the revised manuscript.

Secondly, as suggested by this reviewer, the less sterically hindered methyl alkyl ketone substrates have been tested, such as methyl-cyclohexyl and methyl-cyclopropyl ketones. However, it was very difficult to get crystals of their corresponding reduced alcohol products even after derivatization. Furthermore, the optical rotations of the alcohol products are very small, so that it is also not suitable to predict their absolute configurations by calculations that were based on optical rotations and ECD studies.

5. Compounds C37-C40 are by far the strongest and most interesting evidence in support of the central argument of this paper as, unlike the diaryl ketones in Fig. 2, they cannot be predicted based on trivial visual inspection. However, these 4 examples are not sufficient evidence. For the argument in Figs. 2 and 3 to be compelling, the authors need to provide more examples where alternate, trivial methods of prediction will fail. I would be particularly interested to see examples of ketones where one side is a simple arene and the other is a heterocycle with a local polarizability exceeding that of the arene (so the predicted sense of enantioinduction would be opposite that in Fig. 2 if the compounds are drawn in the same orientation). I also wonder what happens with cyclohexenes (if they are tolerated by the reaction). As mentioned above, it would also be far more compelling if the authors could find a way of demonstrating a quantitative relationship (as they shown in Figure 4) across substrates with widely varied substituents (as they show in Figure 2 and 3).

Our response:

Firstly, we have added six new compounds (C42-C45 and C47-C48) in Fig. 3d for the qualitative correlation study of local polarizability on the sense of enantio-selection, which, together with the existing examples in Figs 2 and 3, could provide strong support for the use of local polarizability-based stereochemical models. These *di*-hetero-aryl ketones in Fig. 3d were structures where applications of alternative, trivial methods would be difficult to predict the absolute stereochemical outcomes. We have also added the corresponding tests in the revised manuscript which were highlighted in yellow: “Lastly, for *di*-hetero-aryl ketones, in particular a range of pyridyl-pyridyl ketones bearing virtually iso-steric hetero-aromatic rings were examined, the absolute configurations of the resulting alcohols C39-C46 were all in accordance with the polarizability-based stereochemical model (Fig. 3d). For alcohols C39-C42 the two pyridyl ring substituents are essentially identical but just differ in their substitution patterns as well as connection positions to the central carbinols, thereby leading to distinctions in their local carbon polarizabilities and subsequent good-to-high levels of product ees. For alcohols C43-C46 in which each structure has two closely comparable pyridyl rings flanked with different substituents, local polarizability mappings allow for straightforward stereochemical predictions that are in good agreement with experimental observations. For benzofuran or benzothiophene-substituted alcohols C47-C48, again a simple and subtle switch of ring substitution patterns would enable sufficient local polarizability difference bringing about the observed enantioselections.”

Figure 3d in the revised manuscript

Secondly, we have attempted to find the ketones that this reviewer is particularly interested in, the ketones where one side is a simple arene and the other is a heterocycle with a local polarizability exceeding that of the arene, however, after multiple attempts, we failed to identify such substrates as heteroatom-substitution induces a decrease in the local polarizability of its α -carbon. We will keep trying our best to find this kind of substrates in the future.

Thirdly, we have run the ATH reactions of similar substrates as cyclohexenes

where alkenes were conjugated with the carbonyl group, however, the alkenes were completely reduced under the reaction conditions.

Finally, it is our ultimate goal to establish quantitative relationships across various substrates as shown in Figs 2 and 3. However, it will be a good start and easier to investigate if only some specific substituted positions varied. Therefore, for the sake of structural comparability, we keep one of the two aryl substituents constant while systematically varying the other ones bearing different substituents. As mentioned above, we have added this quantitative relationship in Figure S49 in the revised Supporting Information.

6. In figure 4, the authors correlate local polarizability with e.e. in 3 different reactions. In contrast to the data in Figs. 2 and 3, the compounds used for Fig. 4 are all simple arenes with varied substituents, making them amenable to classic LFERs such as Hammett plots. The authors actually show Hammett plots for these data in Fig. S48, with the result that correlations comparable to those shown in Fig. 4 are obtained via a straightforward Hammett analysis for all but the Sharpless dihydroxylation (and even then, it seems that only a single substituent, 3,5 dimethoxy benzene, was predicted inaccurately). Given that the Hammett analysis is much easier to perform than computational analysis of local polarizability, it is not clear to me that the value of predicting that one substrate more accurately is worth the added complications of using their approach. I also wonder how the correlation with local polarizability compares to measurement of the polarizability of the entire aryl ring, or other properties that can be generated computationally such as the electrostatic potential at a fixed distance above the ring.

Our response: We really appreciate this reviewer's insightful and valuable comments. Indeed, the experimental Hammett's constants are relatively easy to use for relating reaction rates or stereoselectivity. However, a shortcoming is that the experimental data are limited, for a wide variety of substituents, an effectively computable descriptor would serve as an alternative and has its own advantages such as a more direct physical basis. In addition to the qualitative prediction, we also demonstrate the ability of local polarizability in correlating stereoselectivity by using a set of simple substrates. Furthermore, according to the valuable opinion of the reviewer, we also have evaluated the performances of the polarizability of the entire aryl ring and the electrostatic potential at a fixed distance above the ring on their correlations with stereoselectivity. Taking CBS system as example, the results indicate that the electrostatic potential also has a good linear correlation with stereoselectivity, but the polarizability of the entire aryl ring does not.

Corey-Bakshi-Shibata system

There may be significant advantages to using local polarizability over alternatives (visual inspection, Hammett analysis, other computational approaches) in terms of the ease of the analysis (where visual inspection and Hammett analysis is easier, but other computational approaches may be more challenging), the accuracy of the predictions, or the generality of the approach (this strikes me as the most likely advantage of this approach as Hammett analyses can only easily be performed for meta- and para-substituted phenyl rings), but it is not possible to reach a conclusion either way from the data presented here. The manuscript does provide a handful of examples of this: compounds **C37-C40** and the 3,5-dimethoxystyrene in the SAD, but the vast majority of the examples in the current paper could be more easily predicted by other means. **I would be happy to reconsider a revised version of this manuscript where the authors have addressed these concerns and provide stronger evidence for the utility of models based on local polarizability over other alternatives.**

Overall, we really appreciate this reviewer's insightful comments and valuable suggestions for this manuscript. According to his/her suggestions, we have significantly revised this manuscript in three aspects, and we hope that these revisions can address most of the concerns and questions from this reviewer.

Firstly, the Abstract and Introduction have been re-written to focus on introducing the relationship between polarizability and enantio-selection.

Secondly, we have added six new substrates in Fig 3d for qualitative correlation study of local polarizability on the sense of enantio-selection. These new compounds, together with the existing examples in Fig 3d, are difficult to be predicted of the sense of enantio-selections based on alternative methods, such as visual inspection or Hammett analysis.

Finally, we have demonstrated a good linear free energy relationship between substrates' local polarizabilities and ees in our Ru-catalyzed asymmetric transfer hydrogenation as Figure 49 in the revised Supporting Information.

Reviewer 2.

Key Results:

The authors report that local polarizability of atoms vicinal to a substrate's prochiral center predicts the sense and magnitude of enantioinduction. Remarkably, this trend is recapitulated across diverse catalysts, mechanisms, and transformations. These results provide a compelling model for interpreting low enantioselectivities observed in asymmetric hydrogenation of di-alkyl ketones, especially for substrates with large differences in the steric profiles of the two substituents. The authors make clear that polarizability is the most important, but not the only factor that contributes to enantioinduction (steric effects play a role). The authors note that although the Hammett parameter σ is correlated with enantioselectivity, the free energy of enantioinduction is more correlated with polarizability than σ .

Validity:

The results in this paper show a clear correlation between polarizability and enantioinduction. However, this work also shows a clear correlation between σ and enantioinduction. In the absence of the polarizability data, the title of this paper could have easily been "Sigma matters for enantioinduction". This motivates the question, what other factors that have not been considered might account for enantioinduction? Some obvious candidates include factors that are correlated with local (atom-wise) polarizability, namely attributes of the fragment (the whole substituent attached to the pro-chiral center) including polarizability, quadrupole moment, and dispersion interaction energy. Note that the title of this paper could easily be "Dispersion matters for enantioinduction" and this would fall in line with the work of others (see Schreiner reference below).

Since local polarizability cannot be experimentally measured, we cannot determine their error. Without knowing how much error is associated with the measurement, it is perilous to draw conclusions based on the magnitude of the correlation.

Our response: the reviewer's comments are very insightful. Right now, we honestly have been unable to put forth a clear-cut answer in a conceptual manner. Here we wish to emphasize that the experimental data as well as literature results presented in this manuscript could allow for the recognition of such polarizability effect as a simple and general electronic factor predicting the sense and magnitude of enantio-induction. This effect appears to be penetrating, and it deserves attention from the community. And as wisely pointed out by reviewer 1, there may be an important truth lying at the heart of these data. Within the context, we note further that this polarizability-based rationale is not without precedence, and in fact has its roots in earlier theories of reactivity for organic reactions (We have added the corresponding tests in the first paragraph of the Introduction section in

the revised manuscript, which were highlighted in yellow.). Following this study, we will also investigate if there is potential matching between substrate and catalyst polarizabilities in enantio-control. In short, although we noticed that polarizability appears to be a critical factor, we feel it might be wiser to leave the issue for future research and investigation.

Significance:

This study has the potential to be highly significant, but that potential is not fully clear in the current manuscript. As it stands, the paper provides an impressive collection of correlations that span diverse and important transformations. It appears that there is an important truth lying at the heart of these data. However, there is no piercing insight drawn from these correlations.

There are several reports that note correlations between free-energy of enantioinduction and polarizability (or polarizability per volume). It is concerning that none of these reports are cited in the present manuscript (especially the report of Schreiner, which closely aligns with the results reported here on the Corey-Bakshi-Shibata reduction).

<https://onlinelibrary.wiley.com/doi/full/10.1002/anie.202012760> (figure 12)

<https://pubs.acs.org/doi/10.1021/ja101256v> (discussed in main text, figure in SI, pg S-34)

<https://www.nature.com/articles/nchem.1450> (discussed in main text, figure in SI, pg S-27)

Our response:

We sincerely thank this reviewer for the valuable comments and suggestions. These references have been commented and cited in the introduction section in the revised manuscript: “Jacobsen and co-workers reported an intriguing example that polarizability of the arene of a thiourea catalyst strongly influences the enantioselectivity of the polycyclization of the hydroxylactam⁵⁻⁶. Schreiner and co-workers reported that, with a given catalyst in the Corey-Bakshi-Shibata (CBS) reductions, enantioselectivities increase with the computed substrates’ polarizabilities per volume⁷. The trends in the above were understood based on the fact more polarizable substrates lead to stronger non-covalent interactions (cation- π ⁵⁻⁶ or Londer Dispersion⁷) with the catalyst and thus to higher enantioselectivities.”.

Data and methodology:

The data and methodology are sound.

Analytical approach:

Noting the R² value for two independent variables (sigma vs local polarizability) is not exactly robust, but it is not far from the standards of the field.

Improvements:

This work starts with the problem of challenging substrate classes. It provides a compelling model for why those substrates are challenging, but offers no solutions. Since this work does not offer a solution to achieving high enantioselectivity with this class of substrates, it might be prudent to omit them from the introductory figure.

Our response:

we thank this reviewer's valuable suggestions. The relevant wordings on such challenging substrates are deleted, and the whole introduction section has been re-written.

Clarity and Context:

The authors do a great job of introducing an important problem and clearly presenting compelling data that relates to the problem. There is a lack of clarity in the interpretation of the data. It seems that the authors are essentially noting a correlation and then avoiding any interpretation of the correlation. If this is the case, they should state explicitly that the physical basis of the correlation and the extent to which it applies to other asymmetric catalytic reactions is outside the scope of this work. As it is currently written, the manuscript indirectly suggests that local polarizability is the dominant driver of enantioselectivity in all of asymmetric catalysis (see, for example, the first sentence of the conclusion).

Our response: we thank these very sound comments. An interpretation on the polarizability effect and why this may be so is made in the Introduction section in the revised manuscript, through correlating its root to earlier HSAB and DFT theories developed by Pearson, Parr et, al and their co-workers, where polarizability is shown to be as vital and fundamental as energy itself. Please see the details in the revised manuscript.

References:

The Schreiner paper above should be cited. It would be appropriate to cite other works (like those noted above) that have noted a correlation between enantioselectivity and polarizability.

Our response:

The Schreiner' paper, Jacobsen's papers and Sigman's papers that have noted a correlation between enantioselectivity and polarizability have all been commented and cited in the introduction section in the revised manuscript, which have been highlighted in yellow.

Reviewer 4:

Fumin Chen et al. report an experimental and theoretical study of three enantioselective reactions and suggest that local polarizability differences between two α -Cs of substituents attached to a carbonyl correlate with enantioselectivity. In particular, large and small are not effective, while alkyl (not very polarizable) and aryl (polarizable) are effective in promoting enantioselectivity. The examples all involve aryl alkyl ketones (with very few vs. dialkyl or diaryl for comparison).

The authors report experimental data on Ru-catalyzed asymmetric transfer hydrogenation of unsymmetrical aryl-aryl ketones. The e.e.s. are mostly 75-95%. The aryl-alkyl ketones in Figure 3 are also about the same ee range except for C41 and C42, alkyl alkyl, which are 31% and 66%. For three reactions, Figure 4 shows good correlation between measured $\Delta\Delta G^\ddagger$ and the local polarizability difference of substituents. No error bars are given, but R2 values are given. They do the same for two other reactions.

The authors describe the calculations of local polarizabilities “using the D4 dispersion model” or Grimme. While a few words about how this is done are given, actually the reader does not know how this was done even after attempting to read Grimme’s 19-page paper. Although the SI notes a little more detail, use of MO6-2X but then GFN2-xTB, a simple semiempirical tight-binding method, it will still not be possible to reproduce the polarizabilities in this paper without trial and error testing. Another method with Multiwfn was described also.

Our response: We really appreciate this reviewer’s valuable comments. According to the suggestions, we have added a detailed guide for the calculation of the local polarizabilities in the revised Supporting Information, which would be helpful for the readers to easily reproduce these results.

The paper is impressive in the large amount of data – mostly presented in the SI –and correlations. However, there is no attempt to explain why this works. An energy decomposition analysis of a few cases might help – do the stabilizations resulting from polarization parallel polarizabilities?

Our response:

We really appreciate this reviewer’s insightful comments and suggestions. Indeed, to understand the correlation between local polarizabilities and stereoselectivity is important and indeed challenging. We are trying to understand these findings in detail using different tools including energy decomposition analysis. For the catalytic systems studied in this work, the interaction energies are largely between the reacting fragments that involving bond breaking and formation. As a result, an energy decomposition analysis may not be suitable for evaluating various contributions from the electronics including stabilization by polarization at the α -carbon that is not directly involved in bond breaking and formation. We will keep focused on solving this problem by investigating simple model systems, in order to

explain why this works. Therefore, only some preliminary analyses are provided in Supporting Information, which includes key transition states and distortion/interaction analysis.

In summary, the paper is impressive in (1) the amount of data generated almost without discussion and (2) the discovery that the local polarizability of α -Cs correlates with selectivity. But we do not really know what is meant by local polarizability or why it correlates.

Our response: we really appreciate this reviewer's insightful comments and valuable suggestions. A discussion paragraph has been added into the Introduction section in the revised manuscript. The term "local polarizability" (or its variants such as "local hardness" or "local softness"), its physical definition, and its influence on chemical stability as well as reactivity, had in fact already been clearly defined and examined by a school of pioneers, including notably Pearson, Parr, Yang et.al. But all of the above work had been concentrated on achiral systems, what we disclosed here may be considered as a conceptual stretch when polarizability effect was employed into chiral systems accounting for enantio-induction events. Please see the details in the revised manuscript.

Reviewer 3:

The work is interesting by novelty is lacking the catalyst itself is a chiral molecule, therefore can lead to the enantioselective catalysis. Polarizability in this case is not helping to achieve enantiopurity, the results are highly speculative.

XRD figure have problems in the descriptions, you show them in "Sticks" mode why then are you taking about 50% probability in ellipsoids if you do not show those figures in an ellipsoids mode? What a nonsense!

Our response:

We appreciate this reviewer for pointing this mistake out. This has been corrected in the revised Supporting Information.

Reviewers' Comments:

Reviewer #1:

Remarks to the Author:

The authors appear to have made a significant effort to address my concerns with the previous version of this manuscript. I found it particularly significant that the authors have largely clarified their finding as being a correlative rather than a causative relationship between polarizability and enantioselectivity (though I still think phrasing like "the influence of local polarizability on the sense of enantio-selection" should be changed to something like "the relationship between polarizability" or something like that). They also added a range of experimental data that somewhat strengthens the claim that local polarizability may exhibit a better correlation with enantioselectivity than trivial analysis via visual inspection or more commonly considered LFERs such as Hammett values. I would still like to see more examples of edge cases (e.g. phenyl-3-pyridyl ketone, phenyl-4-pyridyl ketone, 3-pyridyl-4-pyridyl ketone, etc.), but I also understand that collecting those data (especially the crystal structures) may be difficult, and I think the additional experiments the authors have added represent a reasonable compromise provided that their claims are presented carefully. Relatedly, I still worry about whether local polarizability is just a reflection of some underlying property that may be better at predicting enantioselectivity and/or easier to use, but I think that is a question for the broader community to decide, and thus, should not stand in the way of publication. I would, however, like to see a fuller discussion of those points added to the main text as well as some smaller changes, all of which are discussed below.

Line 55-56: I find the authors' statement here to be phrased too strongly – it comes across more like a conclusion than a hypothesis. I think this can be addressed with fairly small changes, but I would suggest something more like "Thus, in agreement with Brewster's suggestion, we hypothesized that the polarizability at the atoms and bonds directly attached to the forming chiral center might be particularly strongly related to enantioinduction."

Line 131-134: I still find the discussion of A34 and A35 relative to A36 to be misleading. All three cases have very small differences in local polarizability, whereas the substrates that give reasonable e.e. all have significant steric variation between the two arenes. I think all that can really be said about these 3 substrates is that when the polarizability difference is small steric differences clearly dominate (as is also seen with C37 and C38). I also think that product C36 should be drawn with a wavy bond rather than a wedge due to the low e.e. and the lack of certainty about the sense of enantioinduction.

Line 152-154: I am not convinced that the results with the benzofuran and benzothiophene isomers aren't due to the presence/absence of an ortho-ring fusion and steric effects (as seems to be the case in C34 and C35).

The authors articulate a thoughtful response to my initial comment (that Hammett values may not be available for any given substrate/substituent, whereas the polarizability could be calculated) in their response to reviewer comments. I think this is an important point and the entire discussion, both that the polarizability and Hammett values are relatively co-variant and the potential advantages of their approach (no reliance on previously tabulated values, though it is necessary to mention that tools have been developed for computing Hammett values) should be included in the main text. It would be ideal to also include the Hammett plots in the main text and compare the results obtained by their methods to those obtained using classical methods.

The location of the Hammett plots in the SI are difficult to find since they are neither located in the most sensible place (close to the corresponding plots vs. local polarizability to allow for easy comparison) nor are they listed in the table of contents. This should be fixed, ideally by showing the Hammett plots in the main text, but failing that, at least by mentioning and discussing them in the main text and making them easier to find in the supporting information.

The comparison of local polarizability to whole ring polarizability and electrostatic potential provided in the response to my comments is interesting and informative, but I didn't see it included in the SI. Since the authors have already performed the calculations and prepared a scheme for the CBS reduction, it should be little work to add it to the supporting information.

The manuscript still has quite a few small typographical and grammar errors that should be fixed. I've noted down the ones I spotted, but the authors should check through the manuscript carefully themselves.

Line 11 – add commas in “chirality, or, handedness”

Line 14 – add comma in “Thus,”

Line 16 – remove “well” in “well capable”

Line 18 – change “the” to “a” in “in a ruthenium-catalyzed”

Line 20 – change “enantio-selections” to “enantio-selectivity”

Line 22-23 – It appears to be a more effective factor..” more effective than what? A comparative phrasing like this needs to state what you are comparing it to.

Line 28 – maybe change “to distort with respect to” to “to distortion under the influence of”

Line 32 – change “react faster” to “typically react faster”

Line 33 – removed “had” from “had suggested”

Line 38 and 39 – remove the “that” from “that break more easily” and “that form more easily”

Line 41 – remove “long”

Line 42 – change “an intriguing example” to “intriguing examples.” Also, reference 6 is not a polycyclization of a hydroxylactam but an indole addition to an episulfonium ion.

Line 47 – change “Londer” to “London”

Line 63 – change “systematic” to “systematically”

Line 63-64 change “take advantage of” to “examined”

Line 79 – change “should” to “could”

Line 129 – change “overwritten” to “overridden”

Line 188 – change “dirct” to “direct”

Reviewer #4:

Remarks to the Author:

The authors have valiantly and politely responded to all of the comments and criticisms from the reviewers. For my section, they added a good discussion of the calculations in the SI. This paper is detailed and lengthy, and the reviews have perhaps exceeded the paper in both these respects. The results are like a ML paper, with some straight lines for ee in this case versus the local polarizability but only a bit of speculation about why. The manuscript is publishable as is.

Reviewer #5:

None

Reviewer 1:

The authors appear to have made a significant effort to address my concerns with the previous version of this manuscript. I found it particularly significant that the authors have largely clarified their finding as being a correlative rather than a causative relationship between polarizability and enantioselectivity (though I still think phrasing like “the influence of local polarizability on the sense of enantio-selection” should be changed to something like “the relationship between polarizability” or something like that). They also added a range of experimental data that somewhat strengthens the claim that local polarizability may exhibit a better correlation with enantioselectivity than trivial analysis via visual inspection or more commonly considered LFERs such as Hammett values. I would still like to see more examples of edge cases (e.g. phenyl-3-pyridyl ketone, phenyl-4-pyridyl ketone, 3-pyridyl-4-pyridyl ketone, etc.), but I also understand that collecting those data (especially the crystal structures) may be difficult, and I think the additional experiments the authors have added represent a reasonable compromise provided that their claims are presented carefully. Relatedly, I still worry about whether local polarizability is just a reflection of some underlying property that may be better at predicting enantioselectivity and/or easier to use, but I think that is a question for the broader community to decide, and thus, should not stand in the way of publication. I would, however, like to see a fuller discussion of those points added to the main text as well as some smaller changes, all of which are discussed below.

Line 55-56: I find the authors’ statement here to be phrased too strongly – it comes across more like a conclusion than a hypothesis. I think this can be addressed with fairly small changes, but I would suggest something more like “Thus, in agreement with Brewster’s suggestion, we hypothesized that the polarizability at the atoms and bonds directly attached to the forming chiral center might be particularly strongly related to enantioinduction.”

Our response: We thank this reviewer for the support and the valuable suggestions. In the revised manuscript, the sentence that this reviewer suggested “Thus, in agreement with Brewster’s suggestion, we hypothesized that the polarizability at the atoms and bonds directly attached to the forming chiral center might be particularly strongly related to enantioinduction.” has replaced the original one.

Line 131-134: I still find the discussion of A34 and A35 relative to A36 to be misleading. All three cases have very small differences in local polarizability, whereas the substrates that give reasonable e.e. all have significant steric variation between the two arenes. I think all that can really be said about these 3 substrates is that when the polarizability difference is small steric differences clearly dominate (as is also seen with C37 and C38). I also think that product C36 should be drawn with a wavy bond rather than a wedge due to the low e.e. and the lack of certainty about the sense of enantioinduction.

Line 152-154: I am not convinced that the results with the benzofuran and

benzothiophene isomers aren't due to the presence/absence of an ortho-ring fusion and steric effects (as seems to be the case in C34 and C35).

Our response: We really appreciate this reviewer's valuable suggestions and comments. We do not rule out the importance of steric effects on the enantio-selection. However, in this work, we want to emphasize that no matter how differences of steric sizes between two sides are, (in some cases where left substituents are more sterically hindered than the right ones, such as C34, C35 that this reviewer mentioned; or in some cases where both sides have exactly the same steric sizes, such as C39, C 40, C47 and C48; or in some cases where the right substituents are more sterically hindered than the left ones, such as C26), the hydroxyl groups facing outward were obtained provided substituents with larger local polarizability were situated at the left and substituents with smaller local polarizability were situated at the right.

In our opinion, the example of A36 perfectly demonstrates the significance of local polarizability: naphthyl rings are obviously more sterically hindered than phenyl rings, however, the ee of C36 is almost zero because of almost the same local polarizabilities on the two arenes.

In order to emphasize the importance of both steric effects and polarizability-based electronic effects, we added one sentence: "Although enantio-selection contribution from conducive steric effect cannot be ruled out, the fact that the α -position of naphthyl ring bears higher electronic densities than those of β -position, thereby more labile for accepting electrophiles, echoes well with the selectivity rules widely observed in aromatic electrophilic substitution reactions." between the contexts of A34-35 and A36 in the revised manuscript.

Furthermore, the wedge bond of C36 has been revised to a wavy bond in the revised manuscript.

The authors articulate a thoughtful response to my initial comment (that Hammett values may not be available for any given substrate/substituent, whereas the polarizability could be calculated) in their response to reviewer comments. I think this is an important point and the entire discussion, both that the polarizability and Hammett values are relatively co-variant and the potential advantages of their approach (no reliance on previously tabulated values, though it is necessary to mention that tools have been developed for computing Hammett values) should be included in the main text. It would be ideal to also include the Hammett plots in the main text and compare the results obtained by their methods to those obtained using classical methods.

The location of the Hammett plots in the SI are difficult to find since they are neither located in the most sensible place (close to the corresponding plots vs. local polarizability to allow for easy comparison) nor are they listed in the table of contents. This should be fixed, ideally by showing the Hammett plots in the main text, but failing that, at least by mentioning and discussing them in the main text and making them easier

to find in the supporting information.

Our response: we appreciate this reviewer's thoughtful suggestions and comments. According to this reviewer's suggestion, the Hammett plots have been put added in the Table of Contents as the section of 3.3.2 in the revised Supplementary Information. Since the main manuscript has already been very long, we would like to put the Hammett plots and the corresponding discussions in the Supplementary Information. We hope this reviewer will understand this.

The comparison of local polarizability to whole ring polarizability and electrostatic potential provided in the response to my comments is interesting and informative, but I didn't see it included in the SI. Since the authors have already performed the calculations and prepared a scheme for the CBS reduction, it should be little work to add it to the supporting information.

Our response: we thank this reviewer for the valuable suggestions. The study of the relationship between the whole ring polarizability and electrostatic potential has been added in the revised Supplementary Information as 3.3.3.

The manuscript still has quite a few small typographical and grammar errors that should be fixed. I've noted down the ones I spotted, but the authors should check through the manuscript carefully themselves.

Our response: we thank this reviewer for the valuable suggestions. We have checked through this manuscript carefully again by ourselves, and all the suggestions below have been revised.

Line 11 – add commas in “chirality, or, handedness”

This has been corrected.

Line 14 – add comma in “Thus,”

This has been corrected.

Line 16 – remove “well” in “well capable”

This has been corrected.

Line 18 – change “the” to “a” in “in a ruthenium-catalyzed”

This has been corrected.

Line 20 – change “enantio-selections” to “enantio-selectivity”

This has been corrected.

Line 22-23 – It appears to be a more effective factor...” more effective than what? A comparative phrasing like this needs to state what you are comparing it to.

This sentence has been removed due to the exceeding words number of the abstract.

Line 28 – maybe change “to distort with respect to” to “to distortion under the influence of”

This has been corrected.

Line 32 – change “react faster” to “typically react faster”

This has been corrected.

Line 33 – removed “had” from “had suggested”

This has been corrected.

Line 38 and 39 – remove the “that” from “that break more easily” and “that form more easily”

This has been corrected.

Line 41 – remove “long”

This has been corrected.

Line 42 – change “an intriguing example” to “intriguing examples.” Also, reference 6 is not a polycyclization of a hydroxylactam but an indole addition to an episulfonium ion.

This has been corrected and the corresponding texts for ref 6 has been added in the sentence: intriguing examples that polarizability of the arene of a thiourea catalyst strongly influences the enantioselectivity of the polycyclization of the hydroxylactams⁵ and the ring-opening of episulfonium ions with indoles⁶

Line 47 – change “Londer” to “London”.

This has been corrected.

Line 63 – change “systematic” to “systematically”

This has been corrected.

Line 63-64 change “take advantage of” to “examined”

This has been corrected.

Line 79 – change “should” to “could”

This has been corrected.

Line 129 – change “overwritten” to “overridden”

This has been corrected.

Line 188 – change “dirct” to “direct”

This has been corrected.